# Targeting the CD40-CD154 Signaling Pathway for Treatment of Autoimmune Arthritis

**DOI:** 10.3390/cells8080927

**Published:** 2019-08-18

**Authors:** Jenn-Haung Lai, Shue-Fen Luo, Ling-Jun Ho

**Affiliations:** 1Division of Allergy, Immunology, and Rheumatology, Department of Internal Medicine, Chang Gung Memorial Hospital, Chang Gung University, Taoyuan 33305, Taiwan; 2Graduate Institute of Medical Science, National Defense Medical Center, Taipei 11490, Taiwan; 3Institute of Cellular and System Medicine, National Health Research Institute, Zhunan 35053, Taiwan

**Keywords:** CD40, CD154, costimulation, autoimmune, arthritis

## Abstract

Full activation of T lymphocytes requires signals from both T cell receptors and costimulatory molecules. In addition to CD28, several T cell molecules could deliver costimulatory signals, including CD154, which primarily interacts with CD40 on B-cells. CD40 is a critical molecule regulating several B-cell functions, such as antibody production, germinal center formation and cellular proliferation. Upregulated expression of CD40 and CD154 occurs in immune effector cells and non-immune cells in different autoimmune diseases. In addition, therapeutic benefits have been observed by blocking the CD40-CD154 interaction in animals with collagen-induced arthritis. Given the therapeutic success of the biologics abatacept, which blocks CD28 costimulation, and rituximab, which deletes B cells in the treatment of autoimmune arthritis, the inhibition of the CD40-CD154 axis has two advantages, namely, attenuating CD154-mediated T cell costimulation and suppressing CD40-mediated B-cell stimulation. Furthermore, blockade of the CD40-CD154 interaction drives the conversion of CD4+ T cells to regulatory T cells that mediate immunosuppression. Currently, several biological products targeting the CD40-CD154 axis have been developed and are undergoing early phase clinical trials with encouraging success in several autoimmune disorders, including autoimmune arthritis. This review addresses the roles of the CD40-CD154 axis in the pathogenesis of autoimmune arthritis and its potential as a therapeutic target.

## 1. Introduction

Patients with autoimmune arthritis, which is mainly comprised of rheumatoid arthritis (RA), psoriatic arthropathy (PsA) and ankylosing spondylitis (AS), suffer from aggressive and long-lasting joint destruction, and many of these patients end up with joint deformity and disability. The process of joint destruction damages cartilage, bone and supporting tissues. Complex immune responses involving immune effector cells and non-immune cells contribute to the whole process of joint inflammation and bony damage, and the underlying mechanism is not completely understood. The initial treatment involves several conventional synthetic disease-modifying antirheumatic drugs (csDMARDs), such as methotrexate (MTX), hydroxychloroquine (HCQ), cyclosporin, sulfasalazine (SSZ) and leflunomide, which nonspecifically inhibit inflammatory reactions to control disease progression and prevent joint damage in patients with autoimmune arthritis [1]. In patients who show an inadequate therapeutic response to csDMARDs, biological DMARDs (bDMARDs) produced from living cells can benefit patients with autoimmune arthritis. In the past few years, new bDMARDs targeting different cytokines, including tumor necrosis factor (TNF), interleukin (IL)-1, IL-6, IL-12/23p40 and IL-17, have been rapidly introduced to the markets. In addition, for the same purposes, small molecule inhibitors targeting membrane-proximity kinases, such as Janus kinase inhibitors, have also been developed in recent years. Unfortunately, there remains a population of patients with autoimmune arthritis who do not respond to these treatments and are suffering from sustained and irreversible joint destruction.

Aside from cytokine-targeting bDMARDs, there are two unique bDMARDs, namely rituximab and abatacept. Rituximab, an anti-CD20 monoclonal antibody (mAb) that specifically targets CD20 molecules on B-cells to delete certain stages of the B cell population during their development. Abatacept, a cytotoxic T-lymphocyte–associated antigen 4 (CTLA-4)-immunoglobulin Fc portion fusion protein, which blocks the T-cell costimulation signal delivered from a costimulatory molecule CD28 to inhibit T cell activation. The activation of T cells plays major roles in the pathogenesis of autoimmune arthritis, and effective suppression of T cell activation serves as a very important and effective approach for immunomodulatory therapy [2,3]. Full activation of T lymphocytes requires two signals: One from the T cell receptor and the other from a costimulatory molecule. Many receptor-ligand interactions have been proposed to provide costimulatory signals for T cell activation, including CD28-B7-1/B7-2, inducible costimulatory molecule-B7-related protein-1, CD70-CD27, CD40-CD154, and OX40-OX40 ligand [4]. The success of the development of abatacept, which inhibits the CD28-B7-1/B7-2 costimulatory signal in RA patients, raises the possibility that other costimulatory pathways may serve as new therapeutic targets for the treatment of patients with autoimmune arthritis. Determination of whether targeting these non-CD28 costimulatory pathways exhibit more benefits or causes fewer adverse events than those in patients receiving abatacept is of interest.

From a therapeutic point of view, among these reported costimulatory pathways, the CD40-CD154 axis is of particular interest. Increased expression of CD154 has been observed on immune effector cells, in addition to T cells, such as B cells and monocytes, and non-immune cells, such as epithelial cells, endothelial cells and fibroblasts in the serum or in inflamed tissues from patients with autoimmune arthritis [5,6,7]. The expression levels of CD154 in CD4+ T lymphocytes correlate well with disease severity, clinical outcomes and disease remission as well as therapeutic response to anti-TNF treatment in RA patients [8,9,10]. In parallel, upregulation of CD40 has been demonstrated in synovial fluid monocytes and articular chondrocytes from RA patients [11,12], and its levels correlate with disease remission after anti-TNF treatment in RA patients [13]. Furthermore, the increase of CD40 has been consistently observed in animal models of collagen-induced arthritis (CIA), and its expression correlates with the upregulated levels of proinflammatory cytokines such as TNF-α and IL-6 and adhesion molecules such as intercellular adhesion molecule 1 (ICAM-1) and vascular cell adhesion molecule 1 (VCAM-1) [14]. Functionally, the engagement of CD40 molecules expressed on synovial fibroblasts from RA patients could induce cellular proliferation, and interferon-gamma (IFN-γ), and to a lesser extent TNF-α, appears to be responsible for upregulating CD40 expression on fibroblasts [15]. Moreover, early studies by Saito et al. showed that the CD28-independent graft-vs-host disease (GVHD) reaction could be significantly attenuated by treatment with anti-CD154 mAb, suggesting the superiority of the CD40-CD154 axis among non-CD28 costimulatory pathways in regulating GVHD-mediated immune responses [16]. All of these findings suggest potential benefits of CD40-CD154 blockade in achieving effective control of inflammation because the inhibition of CD40/CD154 signaling not only decreases T cell costimulation but also inhibits the stimulatory signal to B cells and to other CD40-expressing cells, such as fibroblasts, macrophages, and dendritic cells.

The critical roles of the CD40-CD154 interaction have also been suggested in many immune-mediated disorders, such as atherogenesis-mediated cardiovascular disorders [17], Sjogren’s syndrome [18], systemic sclerosis [19], immune thrombocytopenic purpura [20] and inflammatory bowel disease [21]. To be concise, in the major part of this review, we specifically focus on discussing the potential of targeting the CD40-CD154 costimulatory pathway to develop therapeutics for patients with autoimmune arthritis.

## 2. CD154 and the CD40-CD154 Interaction

Tumor necrosis factor is a crucial cytokine mediating inflammatory reactions, and anti-TNF inhibitors have been generally used to treat patients with autoimmune arthritis who show inadequate therapeutic responses to csDMARDs. There are 19 structurally related soluble or membrane-expressed proteins in the TNF superfamily (TNFSF), and CD40 ligand (CD40L or CD154), also known as TNFSF5, is one of them [22]. CD154 is a type II membrane glycoprotein with a molecular weight of 33 kDa and is transiently expressed on the surface of activated CD4+ T cells, interacting with its cognate receptor CD40 on B cells [23,24]. A subgroup of T cells, T follicular helper cells, mediate important cell-cell interactions with B cells occurring within follicles of secondary lymphoid organs and stimulate and govern B cells to produce antibodies [25]. After interacting with CD40 through CD154, T cells become activated and produce a variety of cytokines and regulate proinflammatory immune responses [26]. Meanwhile, the interaction with CD40 also leads CD154 to being cleaved by the matrix metalloproteinases a disintegrin and metalloproteinase domain-containing protein (ADAM)10 and ADAM17 and released from the cell surface as a truncated 18-kDa protein [27]. While circulating in the blood stream, this truncated protein may be present as monomer, dimer or trimer or form multimeric complexes with the membrane-bound CD154 on the cell surface [28,29,30]. The formation of a higher multimeric organization may enhance the biological activity of CD154, and compared to the membrane-bound CD154, the soluble form of CD154 shows lower biological activity [31,32].

In addition to the close contact between T cells and B cells in lymphoid organs, the cell-cell interaction delivering the CD40-CD154 signal also occurs in peripheral inflamed tissues, such as joints [33]. T cells activated with anti-CD3 or cytokines induced the release of monocyte chemotactic protein-1, IL-8, and IL-6 from endothelial cells through cell-cell contacts, and such events were blocked by exogenously added anti-CD154 antibody [34]. Fibroblast-like synovial cells cocultured with CD40L+ L cells produced vascular endothelial growth factor, and the effects could not be inhibited by the treatment with neutralizing Abs to IL-1β, TNF-α, or transforming growth factor-β [35]. This study suggests a unique role of the CD40-CD154 interaction that is unresponsive to inhibition by critical proinflammatory cytokines, such as TNF-α and IL-1β, in immune responses. However, analyzing gene signatures revealed that many CD154-responsive genes detected in immature dendritic cells and naive B cells were also reproducibly observed in synovial tissues from patients with undifferentiated arthritis, early RA, and established RA [36]. The results suggest that CD154-responsive stimulatory signals are shared by different subsets of autoimmune arthritis. Interestingly, the cognate receptor for CD154 is not limited to CD40, and several molecules are considered capable of binding to CD154, such as αIIbβ3, α5β1 (VLA-5) and αMβ2 (Mac-1) integrins, indicating that CD154 is a possible target for the treatment of atherosclerosis [24]. Together with many immune effector cells and non-immune cells, through cell-cell contacts and secreted molecules such as cytokines and inflammation-related factors, activated T cells via the CD40-CD154 interaction play pivotal roles in regulating joint damage in direct and indirect manners.

## 3. CD40 and CD40-Mediated Signaling

B cells play critical roles in the pathogenesis of RA, a notion also supported by the success of the B-cell-depleting agent rituximab in the treatment of RA patients. As one of the hallmarks of this disease, serum levels of anti-citrullinated protein autoantibodies and rheumatoid factor from activated B cells/plasma cells are highly correlated with the extraarticular manifestations, prognosis and disease severity of RA patients [37]. Similar to T cells, the activation of B cells also requires signals from both the antigen receptor and a costimulatory molecule [38]. CD40, a costimulatory molecule glycoprotein with 277 amino acids, also known as TNFRSF5, was originally identified as a receptor on B-cells and was later found to be expressed in various immune effector cells, such as monocytes, macrophages and dendritic cells, as well as activated T cells and non-immune cells, such as vascular endothelial cells, epithelial cells and fibroblasts [39,40,41,42]. In addition, the expression of CD40 can be widely detected on the cell surface of many hematopoietic cells, such as mast cells, basophils, natural killer (NK) cells, and platelets, suggesting the extensive involvement of this molecule in various immune responses [43,44]. After interaction with its ligand CD154, CD40 is internalized into the cell, and sequential signal transductions are initiated, leading to several crucial signaling events, such as stimulation of antibody production and class switching, prevention of cellular apoptosis, formation of germinal center in lymph nodes, more efficient antigen presentation by upregulating the related receptors, production of cytokines, and cellular proliferation and differentiation as well as the generation of memory B cells and plasma cells [45,46,47].

Because the cytoplasmic domain of CD40 does not contain enzymatic activity, it recruits and interacts with adapter proteins, termed TNF-receptor-associated factors (TRAFs), at the cytoplasmic tail and triggers downstream signaling pathways. There are seven TRAFs, and each contains a conserved TRAF domain (except for TRAF7), a region forming a coiled-coil domain responsible for homo- or hetero-oligomerization of TRAF in the N-terminus, and seven to eight anti-parallel β-strand folds responsible for TRAF recruitment to the cytoplasmic tail of TNF receptors in the C-terminus [48,49]. There are several CD40-interacting TRAFs, including TRAF1, TRAF2, TRAF3, TRAF5, and TRAF6 [50], although TRAF5 appears to have a weaker association with CD40 compared to the viral oncogenic CD40 mimic latent membrane protein 1 [51]. To date, no reports have suggested a direct interaction between CD40 and TRAF4 or TRAF7 [52]. In addition to CD40, many receptors and intracellular signaling molecules have been shown to interact with TRAF family proteins, including CD30, Ox40, TRADD, LMP1, TNFR2, RANK, IRAK, RIP2, GPIb, GPVI, and TANK, and mediate diverse signaling events involved in various immune responses [49].

Following the interaction between CD40 and TRAFs, one of the most important downstream signaling events is the activation of the NF-κB signaling pathway (Figure 1). The NF-κB signaling pathway has been generally recognized as a critical player contributing to a variety of systems in maintaining homeostasis and in pathogenic processes, including cell proliferation, cell death and inflammatory reactions. The activation of the canonical NF-κB (NF-κB1) pathway is dependent on the phosphorylation and degradation of IκBα regulated by NF-κB-inducing kinase (NIK) and the translocation of transcription factors such as p50/RelA or p50/c-Rel heterodimers from cytosol into the nucleus. The activation of the canonical NF-κB pathway occurs quickly, and there is no need to resynthesize proteins during the process. In comparison, the activation of the noncanonical NF-κB (NF-κB2) pathway occurs slower and requires new protein synthesis, and the processing of p100 leads to nuclear localization of mainly p52/RelB heterodimers [53,54]. Stimulation of CD40 activates both canonical NF-κB (NF-κB1) and noncanonical NF-κB (NF-κB2) signaling pathways. CD40 ligation of RA synovial fibroblasts by activated T cells activates, in addition to the NF-κB signaling pathway, extracellular-signal-regulated kinase (ERK)-1/2 and p38 mitogen-activated protein kinase (MAPK), upregulates receptor activator of NF-κB ligand (RANKL) expression and enhances osteoclastogenesis [55]. Furthermore, CD40 stimulation leads to the activation of phosphoinositide 3-kinase (PI3K) and phospholipase Cγ signaling pathways [56]. Given the diverse expression of CD40 in different immune effector cells and non-immune cells, the CD40-mediated effects can be different. For example, CD40 activation induced expression of the adhesion molecules VCAM-1, ICAM-1, and E-selectin on endothelial cells and fibroblasts [44]; however, CD40 signaling inhibited T cell activation and the release of proinflammatory cytokines as well as inflammation of adipose tissue in mice [57].

Early studies from Pullen et al. revealed the presence of core binding sites for TRAF1, TRAF2, and TRAF3 in CD40 molecules and suggested that the integrated signals from individual TRAFs together contributed to the overall CD40-mediated responses [58]. In contrast to other TRAF family proteins, TRAF1 did not contain the N-terminal ring finger motif, and its expression is limited to the spleen, lung and testis according to Northern blot analysis [48]. Most TRAF1 proteins form a heterotrimer with TRAF2 to bind to CD40, and in the absence of TRAF1, there is increased degradation of both TRAF2 and TRAF3 in response to CD40 stimulation [59]. The results suggested that TRAF1 plays important roles in maintaining the level and function of both TRAF2 and TRAF3 [59]. In addition, stimulation of CD40 induced apoptosis of dendritic cells with a deficiency of TRAF1 [60]. Regarding the effects on downstream signaling molecules, Xie et al. showed that the deficiency of both TRAF1 and TRAF2 nearly completely blocked CD40-induced IκBα degradation [59]. Accordingly, there was no detectable level of p52 or RelB in the nucleus in B cells with deficiency of both TRAF1 and TRAF2 in response to anti-CD40 stimulation [59]. In parallel, stimulation with anti-CD40 mAb induced activation of both JNK1 and JNK2, and the effects were completely inhibited in B cells with deficiency of both TRAF1 and TRAF2 [59]. Notably, researchers have reported different conclusions when examining the roles of TRAF1 in the activation of NF-κB signaling in different tissue cells. For example, the enhanced expression of TRAF1 by a genetic approach in TNF-treated HEK293T cells may cause negative effects [61,62] or positive effects [63] in TNFR family-induced NF-κB activation. The interesting roles of TRAF1 have been comprehensively reviewed by Edilova et al. [64].

Rowland et al. reported that deficiency of either TRAF2 or TRAF6 alone in mouse B cell lines did not result in a significant defect in CD40-mediated NF-κB2 activation; however, a combined deficiency of both TRAF2 and TRAF6 resulted in profound suppression of CD40-mediated NF-κB2 activation [65]. In the absence of a molecule, Tespa1, the CD40 stimulation-mediated stabilization of TRAF6, but not TRAF2 or TRAF3, was defective [66]. The activation of NF-κB required the formation of a TRAF6-p62 complex, and the interruption of this complex formation did not affect TRAF6-independent signaling pathways but led to the impairment of NF-κB activation in macrophages [67]. Specific inhibition of TRAF6 with recombinant high-density lipoprotein (rHDL) nanoparticles without affecting CD40-TRAF2/3/5 interactions resulted in decreased expression of CD40 and β2-integrin as well as reduced activation and migration of macrophages. Meanwhile, this treatment did not show any effect on T-cell proliferation and costimulation, immunoglobulin isotype switching, or germinal center formation [68].

Although the integrated signal from individual TRAFs provides positive effects for CD40-mediated proinflammatory reactions, TRAF3 appears to play a negative role in CD40-mediated signaling. TRAF3 loss-of-function mutations are commonly observed in B-cell lymphoma [69]. TRAF3 is constitutively associated with NIK, which is also an important regulator of the noncanonical/NF-κB2 pathway induced by IL-1 or TNF superfamily members [70,71], in a dynamic manner, and the TRAF3-NIK interaction leads to NIK degradation by the proteasome [72]. Stimulation with both anti-CD40 and BAFF caused persistent degradation of TRAF3 and increased stability, reversed the blockage of NIK and resulted in activation of the downstream NF-κB signaling molecules [73]. Because TRAF3 knockout results in neonatal death, a conditional B cell-specific knockout of TRAF3 was generated in mice to address TRAF3-related questions. The specific knockout of the *Traf3* gene in B cells of mice activated the noncanonical NF-κB signaling pathway resulting from constitutive p100 processing and increased expression of p52 and Rel B in the nucleus [74,75]. Interestingly, TRAF3 also regulates B cell metabolism by functioning as a resident nuclear protein via association with the transcriptional regulator cAMP response element binding protein (CREB) and Mcl-1, the antiapoptotic target of CREB [76,77]. Collectively, these findings suggest a tight regulation and interaction between TRAFs and CD40 as well as the non-overlapping functions of individual TRAFs.

## 4. The CD40-CD154 Interaction in the Pathogenesis of Autoimmune Disorders

The significance of the CD40-CD154 interaction in autoimmune disorders was investigated by using a neutralizing mAb or RNA interference. Early et al. reported that treatment with anti-CD154 mAb effectively reduced anti-DNA autoantibody production, improved renal disease and significantly prolonged survival in New Zealand Black (NZB) x New Zealand White (NZW) lupus-prone mice [78]. Amazingly, the therapeutic benefits in controlling lupus nephritis severity and reducing lupus nephritis incidence appeared to be sustainable, and the effect lasted even long after the anti-CD154 antibody had been cleared from the mice [79]. Treatment with a rat/mouse chimeric anti-mouse CD40 mAb in NZB/W-F1 mice after the onset of severe proteinuria could reverse the already established nephritis with severe proteinuria and recover the disease status back to normal glomerular and tubular morphology [80]. The therapeutic benefits were confirmed by analyzing genes associated with proteinuria and the damage of renal parenchymal cells. By examining a different strain of mice, MRL/Mp-lpr/lpr, the authors reproducibly observed the therapeutic effects of anti-CD40 treatment, and the therapeutic benefits were even extended to include improvement in salivary gland function and alleviation of joint inflammation [80].

In a disease model of mice with CIA, the introduction of CD40 siRNA resulted in a significant reduction in disease severity, and the effects could be demonstrated in both pre- and post-immunization manners [81]. The therapeutic effects could also be reflected in a decrease in proinflammatory cytokine production and antibody production and the upregulation of regulatory T cells (Tregs) [81]. Similar observations were also demonstrated in studies of anti-CD154 mAb treatment, which resulted in the reduction of joint inflammation and erosion of cartilage and bone in CIA mice [82]. In contrast, the introduction of stimulatory anti-CD40 mAb induced the production of collagen II-specific IgG2a antibodies and increased interferon-gamma (IFN-γ) production, causing earlier onset and more severe disease in mice with CIA [83]. In a disease model with CIA in monkeys, the introduction of anti-CD154 mAb improved arthritis symptoms and movement, decreased the numbers of proliferating B cells and reduced the CD4+/CD8+ cell ratio in peripheral blood [84]. In addition to the reduction of cartilage damage, therapeutic effects were also observed in the non-progression of obscurity of the epiphysis and the surroundings in anti-CD154-treated animals by radiographic examination. Unexpectedly, this treatment also resulted in a significant reduction in hemoglobin concentrations (from 11.78  ±  1.27 g/dL to 7.84  ±  0.83 g/dL at week 16 post treatment). A reduction in platelet count was also observed in some anti-CD154-treated monkeys [84].

The effects of CD154 blockade were examined in a mouse model of antigen-specific mixed chimerism. In this study, the authors demonstrated that by reducing the reactive T cell response through CD154 blockade, the secretion of proinflammatory cytokines such as IL-6, IL-1β, TNF, and IL-12 from antigen-presenting cells was reduced [85]. Notably, this treatment did not affect the expression of MHC and costimulatory molecules on antigen-presenting cells [85]. Aside from the inhibition of the CD154-mediated T cell costimulation signal and CD40-mediated activation signal to B cells and antigen-presenting cells by CD40/CD154 blockade, anti-CD154 mAb treatment also induced antigen-specific CD4+CD25+FoxP3+ Tregs [86]. Examining an animal model of heart transplantation, Warren et al. further identified the localization of these Tregs into specific areas in the draining lymph nodes of heart allografts [87]. A CD154 neutralizing antibody, MR1, in addition to inhibiting inflammatory responses, induced more bone formation and increased trabecular bone mass in the spine. The underlying mechanisms involve increased Treg development and elevated expression of CTLA-4 leading to inhibition of the CD28 costimulatory signaling pathway and the expression of the bone anabolic ligand Wnt-10b in CD8+ T cells [88].

## 5. Clinical Analyses of Blocking the CD40-CD154 Axis in Humans

Although there are certain similarities in disease pathogenesis and clinical manifestations in different autoimmune disorders, such as SLE and RA, the successful treatment of SLE patients with certain drugs may not translate to their useful therapeutics for RA patients. In other words, the potential success of anti-CD40 or anti-CD154 in patients with SLE may not necessarily indicate its success in patients with RA. For example, the accumulated evidence thus far suggests that anti-TNF agents may have limited or no role in the treatment of SLE, although they showed success in RA patients. However, the biologic rituximab targeting CD20 molecules on B cells is effective in RA patients and may also have potential in SLE patients with severe and refractory lupus nephritis [89]. Similar examples can be found in treatment with csDMARDs, such as HCQ and MTX, in both SLE patients and RA patients. Therefore, the findings and experience with anti-CD40/CD154 clinical trials studied in patients with different autoimmune diseases, not limited to autoimmune arthritis, remain valuable for obtaining more solid information regarding the efficacy, mechanism and safety issues of anti-CD40/CD154 treatment in humans. Table 1 shows the up-to-date knowledge about some representative products targeting the CD40-CD154 axis for therapeutics of autoimmune disorders, including autoimmune arthritis. The authors also recommend the excellent review by Karnell et al. [90].

### 5.1. Anti-CD154

A phase I study examining the safety and pharmacology of a humanized anti-CD154 mAb (IDEC-131) enrolling 23 symptomatic SLE patients was conducted. The patients were allocated to cohorts of 3 to 5 patients with each receiving a single dose of 0.05, 0.25, 1.0, 5.0, or 15.0 mg/kg of IDEC-131 intravenously in a dose-escalating manner. The drug safety in patients was followed for 3 months. The results suggested that IDEC-131 was safe and well tolerated [91]. Subsequently, a phase II, double-blind, placebo-controlled study enrolling 85 patients with mild-to-moderately active SLE was investigated. The patients received 6 infusions with placebo or IDEC-131 at concentrations ranging from 2.5 mg/kg to 10.0 mg/kg over a 16-week period. The primary endpoint was examining the changes in the Systemic Lupus Erythematosus Disease Activity Index (SLEDAI) at week 20, and safety was evaluated through week 28 by measuring both clinical and laboratory parameters. Although the drug appeared to be well tolerated and there were no safety concerns in the observed period, the results did not show any significant clinical benefits with IDEC-131 treatment compared to the baseline in all treated patients [92]. The therapeutic effects and possible adverse events of a humanized C154 mAb, BG9588, administered at a dosage of 20 mg/kg at biweekly intervals for the first 3 doses and at monthly intervals for 4 additional doses, were evaluated. The study was an open-label, multiple-dose, phase II study enrolling 28 SLE patients with proliferative lupus nephritis. The primary endpoint was the ratio of patients achieving a 50% reduction in proteinuria without worsening of renal function. As indicated, the treatment resulted in significant improvement in several disease activity parameters, such as the production of anti-double stranded DNA antibody, hematuria, the concentration of complement 3 and renal function. Unfortunately, the study was prematurely terminated because of the increased incidence of thromboembolic events, with two patients developing myocardial infarction [93]. As also indicated in the report by the authors, the adverse reaction with thromboembolic events was noted in other BG9588 protocols. Furthermore, the administration of an anti-CD154 mAb, ruplizumab, in patients with lupus nephritis was associated with life-threatening prothrombotic events, although the drug treatment appeared to show a favorable therapeutic response in both serological analysis and renal function measurement of the patients [100]. Given that the introduction of anti-CD154 mAb may be associated with thromboembolic events, the presence of anti-CD154 autoantibodies in patients with SLE was associated with thrombocytopenia but not with thromboembolism [101]. The level of anti-CD154 antibody likely affects the occurrence of thromboembolism.

One of the major concerns of applying anti-CD154 mAb treatment was the occurrence of thromboembolic events that are likely the consequence of crosslinking CD154 expressed on platelets by antibodies preserving the Fc portion. To avoid these fatal adverse events, several approaches were developed and examined in animals, such as the development of aglycosyl anti-CD154 mAbs with reduced binding to the Fcγ receptor and complement [102], a CD154 domain Ab with an inert Fc tail [103] and a high affinity PEGylated monovalent Fab’ anti-CD154 antibody fragment (CDP7657) [104]. All of these listed products were shown to have differential benefits on disease activity in the absence of increased risk of thromboembolic events in mouse models of SLE.

Following the success of newly designed products in animal studies, an anti-CD154 mAb Fab portion conjugated to polyethylene glycol, dapirolizumab pegol, was evaluated in a randomized, double-blind, phase I trial in patients with SLE. Sixteen patients were randomized to receive 30 mg/kg dapirolizumab pegol intravenously followed by 15 mg/kg every 2 weeks for 10 weeks, and eight patients were allocated to receive placebo treatment. The study confirmed the safety and tolerance of the drug with no serious adverse events, no thromboembolic events and no death [94]. Compared to placebo treatment with a 14% response rate (1/7) measured using British Isles Lupus Assessment Group-based Composite Lupus Assessment response, dapirolizumab pegol treatment led to higher response rates (46%, 5/11) in patients with high disease activity at baseline [94,95]. A significant improvement in SLE Responder Index-4 responded by week 12 was also detected (5/12 (42%) in the dapirolizumab pegol group vs 1/7 (14%) in the placebo group). Unfortunately, as described in a press release, further analyses that involved enrollment of more SLE patients with moderate to severe disease activity in a randomized, double-blind, placebo-controlled, phase IIb study did not meet the primary endpoint at 24-week treatment with dapirolizumab pegol [96].

A small protein scaffold (approximately 90 amino acids), Tn3, preserving immunoglobulin-like folds fused to serum albumin to delay its clearance, binds specifically to CD154 and is named VIB4920. The results from preclinical analysis revealed that VIB4920 could inhibit the activation and differentiation of B cells and did not cause platelet aggregation in vitro. After preliminary confirmation of the safety of VIB4920 in healthy volunteers, 57 RA patients were enrolled in a phase Ib clinical trial and allocated to receive either placebo (*n* = 15) or multiple ascending doses of VIB4920 (n = 42) encompassing the dosages of 75, 500, 100 and 1500 mg intravenously, and many clinical and laboratorial parameters were determined to analyze the effectiveness and safety of the drug. The results showed that treatment with VIB4920 significantly reduced disease activity, and more than 50% of the patients receiving two higher doses achieved low disease activity or clinical remission at week 12 [97]. Such a therapeutic benefit was also reflected by the decrease in rheumatoid factor levels and Vectra DA biomarker score, a measurement of 12 biomarkers associated with RA disease activity, in a dose-dependent manner [97]. The patients showed good tolerance of the drug, and there was no specific safety concern throughout the trial.

### 5.2. Anti-CD40

CFZ533, a humanized anti-CD40 IgG1 mAb with a modified Fc portion resulting in a lack of Fcγ-dependent effects, was examined in a phase IIa randomized trial for patients with Sjögren’s syndrome. Compared to those patients receiving placebo treatment, there was a significant improvement in patient-reported indexes and disease activity by measuring EULAR Sjögren’s syndrome disease activity index in patients receiving higher doses of CFZ533 treatment (10 mg/kg) [18,98]. The drug appeared to be well tolerated. CFZ533 has also been evaluated in a randomized, double-blind, placebo-controlled trial for the evaluation of drug safety, pharmacokinetics, and pharmacodynamics (ClinicalTrials.gov. 2017. https://clinicaltrials.gov/ct2/show/NCT02089087). Both healthy subjects and RA patients with a total of 75 participants were enrolled in this trial. The trial was completed in Feb. 2017. No further information about the results of this study could be retrieved before submission of this report.

A humanized anti-CD40 mAb, BI 655064, was examined for safety, pharmacokinetics and pharmacodynamics in two different phase I studies. The first study enrolled 72 healthy volunteers who received a single intravenous (0.2–120 mg) or subcutaneous (40–120 mg) dose of BI 655064 or placebo. The drug was well tolerated up to 120 mg intravenously or subcutaneously and did not reveal significant adverse events [105]. Another phase I clinical trial examined the safety, tolerability, pharmacokinetics, and pharmacodynamics of repeated once-weekly BI 655064 subcutaneously over 4 weeks in a multiple-dose study in healthy subjects (*n* = 40). Four doses, including 80, 120, 180, and 240 mg, were examined. No serious adverse events with BI 655064 compared to placebo were observed [106]. A randomized, double-blind, placebo-controlled, phase IIa study on BI 655064 enrolling 67 RA patients with an inadequate therapeutic response to MTX was conducted. The patients were randomized to receive 120 mg BI 655064 (*n* = 44) or placebo (*n* = 23) subcutaneously at a weekly interval for 12 weeks. The primary endpoint was designed to evaluate the proportion of patients achieving 20% improvement in the American College of Rheumatology criteria at week 12. The results revealed that although the BI 655064-treated group had an evident reduction in clinical and biological parameters, including a decrease in activated B-cells, autoantibody production, and inflammatory and bone resorption markers, as well as a favorable safety profile, the significant clinical efficacy was not achieved statistically [99]. Evidently, the ongoing trials with BI 655064 need to recruit more patients to evaluate both efficacy and safety in patients with RA.

## 6. Perspectives

Although significant advancements have been made in drug development for autoimmune arthritis in recent years, some patients remain to show inadequate therapeutic responses to the currently available csDMARDs and bDMARDs. Furthermore, with many different coexisting medical conditions, some bDMARDs appear to be inappropriate for use in treating patients with specific medical needs [107]. Because both the activation of the CD154-mediated costimulatory pathway in T cells and the stimulation of the CD40-mediated signaling pathway in B cells are critical in the pathogenesis of autoimmune arthritis, the introduction of CD40-CD154 blockade provides a wonderful opportunity for therapeutics of these disorders. Thus far, the clinical evaluation of the efficacy and safety of CD40-CD154 blockers involved only limited patient numbers, and more large-scale clinical trials enrolling more patients would hopefully elucidate the potential of blocking CD40-CD154 interaction for the treatment of patients with autoimmune arthritis. As expected, treatment of autoimmune arthritis will still be a major challenge for scientific communities in the future. For these challenges, in addition to testing more new potential therapeutic targets, such as targeting costimulatory molecules on T or B cells, more work to comprehensively elucidate the pathogenesis of autoimmune arthritis is required.

To address the high risk of thromboembolism in patients receiving anti-CD154 blocking antibody treatment, in addition to the introduction of several different anti-CD154 approaches currently under early phases of clinical trials, there remain some products that have not been clinically examined. An approach to use an anti-CD154 chimeric antibody by modifying the human IgG4 Fc portion named the RD-05 antibody was developed. This new product effectively and completely inhibited the CD40-CD154 interaction and T cell-dependent B cell activation in the absence of increasing thromboembolism risk in human FcγRIIA-transgenic mice [108]. A chimeric CD40-specific monoclonal antibody (Chi220) was generated, and its effects on protecting islet xenografts were evaluated. The results suggested that Chi220 improved xenoislet engraftment and survival, and no thromboembolic phenomena were detected [109]. In contrast to antibodies to block CD40-CD154 interactions, a small molecule inhibitor, BIO8898, that inhibits soluble CD154 binding to CD40-Ig could inhibit CD154-dependent apoptosis in vitro. BIO8898 appears to intercalate between two subunits of this homotrimeric molecule and interfere with the protein’s 3-fold symmetry [110]. Another fully human anti-CD40 monoclonal recombinant IgG4, bleselumab, that failed to demonstrate efficacy in a clinical trial of 60 patients with moderate-to-severe psoriasis [111] was assessed alone or in combination with tacrolimus or mycophenolate mofetil, and appeared to provide encouraging benefits as an antirejection agent for improving renal allograft survival [112] and prolonging pancreatic islet allograft survival in cynomolgus monkeys [113]. All of these regimens may be useful for clinical evaluations of autoimmune arthritis in the future.

The X-linked forms of mutations of CD154 affect the folding and stability of the molecule and cause severe immunodeficiency (known as hyper-IgM syndrome) with the manifestations of increased IgM and low or absence of IgG, IgA and IgE in serum [114,115,116], presentations also found in mice with deficiency of CD154 by gene targeting [117,118]. A genetic defect in CD40, an autosomal recessive form, or its downstream signaling molecules such as activation-induced cytidine deaminase and uracil-DNA glycosylase, resulting in defects in CD40-mediated signal transduction in B cells, may also cause hyper IgM syndrome [119,120,121]. One concern is that the overwhelming suppression of CD40-CD154 interaction may result in presentations reminiscent of hyper IgM syndrome. Luckily, these symptoms did not occur in clinical trials with inhibition of the CD40-CD154 interaction. Another concern is that the blockage of the CD40-CD154 interaction may lead to unwanted effects in cancer patients given that treatment with CD40 agonists is beneficial for cancer patients [122]. Accordingly, more clinical experience remains needed in examining the long-term safety of blocking CD40-CD154 signaling pathways for treatment of autoimmune disorders.

## Figures and Tables

**Figure 1 cells-08-00927-f001:**
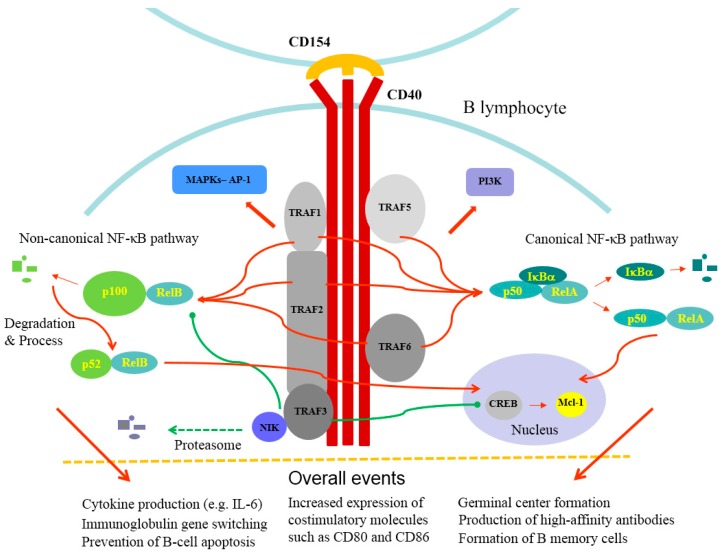
After interacting with CD154, CD40 recruits and interacts with several adapter proteins, including TRAF1, 2, 3, 5 and 6. The accumulated signaling of TRAFs leads to the activation of canonical NF-κB and noncanonical NF-κB signaling pathways. The noncanonical NF-κB signaling pathway involves the processing of p100 and the nuclear translocation of the p52 and RelB heterodimers. TRAF3 is constitutively associated with NIK, leading to its degradation by the proteasome, and CD40 signaling causes TRAF3 degradation and activation of NIK as well as the downstream noncanonical NF-κB signaling pathway. TRAF3 also functions as a resident nuclear protein via association with the transcriptional regulator CREB and Mcl-1. In comparison, the activation of the canonical NF-κB signaling pathway involves the degradation of IκBα and the nuclear translocation of the p50 and RelA heterodimers. Overall, CD40 signaling activates MAPKs and PI3K. The activation of the CD154-CD40 axis results in several events, including cytokine production, immunoglobulin gene switching, prevention of B-cell apoptosis, increased expression of costimulatory molecules such as CD80 and CD86, germinal center formation, production of high-affinity antibodies and formation of B memory cells. TRAF, TNF receptor (TNFR)-associated factor; NF-κB, nuclear factor-kappaB; IκBα, inhibitor of NF-κB; CREB, cAMP response element binding protein; MAPK, mitogen-activated protein kinase, PI3K, phosphoinositide 3-kinases.

**Table 1 cells-08-00927-t001:** Clinical experience with pivotal CD40-CD154 blockers for patients with rheumatic autoimmune disorders.

Aimed Target	Drugs	Structure	Aimed Diseases	Trial Designs	Patient No./Study Period	Results and Comments	Ref.
CD154	IDEC-131	Humanized mAb	SLE *	Phase I & Phase II, DB, PC	23/3 months and 85/7 months	Safe and well-tolerated and no clinical benefits compared to baseline	[91,92]
CD154	BG9588	Humanized mAb	SLE (PLN)	phase II, open-label	28/22–24 wks	Early termination of the trial due to thromboembolic AE although clinical benefits noted	[93]
CD154	Dapirolizumab pegol	Fab	SLE	Phase I, DB & Phase IIb, DB, PC	16/12 wks and 182/24 wks	Clinical benefits in high disease activity group at baseline & Fail to meet the primary endpoint (noted in press release)	[94,95,96]
CD154	VIB4920 (MEDI4920)	Tn3 fusion protein	RA	Phase Ib, DB, PC	57/12 wks	Significant benefits in clinical and laboratorial assessment	[97]
CD40	CFZ533	Humanized mAb	SS; RA	Phase IIa, DB, PC & Phase I, DB, PC	44/12 wks and 75/20 wks (estimated)	Clinical benefits in SS patients & The drug was well-tolerated in RA patients	[18,98], NCT02-089087
CD40	BI 655064	Humanized mAb	RA, fail with MTX	Phase IIa, DB, PC	67/12 wks	Improvement in laboratorial parameters but no significant clinical efficacy	[99]

* Abbreviations: SLE, systemic lupus erythematosus; PLN, proliferative lupus nephritis; RA, rheumatoid arthritis; SS, Sjögren’s syndrome; mAb, monoclonal antibody; Fab, Fab portion of antibody; MTX, methotrexate. DB, double-blinded; PC, placebo-controlled; wks: weeks.

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
