# Peer review of "Targeting the CD40-CD154 Signaling Pathway for Treatment of Autoimmune Arthritis"

_cells, 2019, doi:10.3390/cells8080927_

Round 1
Reviewer 1 Report
This review is excellent in detailed information about CD40-CD154 signaling pathway and the past and ongoing clinical trials to modulate this pathway for autoimmune diseases.
There are several minor issues such as spelling and typographical errors.
In line 100, it is better to add CD40 ligand (CD40L) as synonym since it is a popular word for CD154. In line 120, beta should be changed to β. In line 197, CD4 should be CD40. In line 237, "B cell cancer" is not right word since cancer is used for malignant epithelial tumor. Simply, B cell lymphoma is better. In line 239, there is unnecessary space between NF- and kB2 In line 241, Baff should be BAFF. In line 285, IL-1b should be IL-1β. In Table1, in results and comments of BG9588, trail should be trial.Author Response
Response to Reviewer #1
This review is excellent in detailed information about CD40-CD154 signaling pathway and the past and ongoing clinical trials to modulate this pathway for autoimmune diseases. There are several minor issues such as spelling and typographical errors.
In line 100, it is better to add CD40 ligand (CD40L) as synonym since it is a popular word for CD154. In line 120, beta should be changed to β. In line 197, CD4 should be CD40. In line 237, "B cell cancer" is not right word since cancer is used for malignant epithelial tumor. Simply, B cell lymphoma is better. In line 239, there is unnecessary space between NF- and kB2 In line 241, Baff should be BAFF. In line 285, IL-1b should be IL-1β. In Table1, in results and comments of BG9588, trail should be trial.
Response: We thank the Reviewer very encouraging and helpful comments on our manuscript. All the critiques and typographical errors as indicated above have been corrected as shown in red color in the revised manuscript.
We authors thank again the reviewer very helpful comments on our report.
Reviewer 2 Report
Lai et al summarize current knowledge about CD40-CD154 signaling in arthritis. To this end they explain cellular signaling pathways, summarize knowledge about murine models and clinical experience with CD40-CD154 blockers. Authors summarize a very impressive and huge amount of data with regard to this topic. However at some points they fail to really discuss the data and to come to conclusions or to generate hypothesis. In several paragraphs they only list recent studies and jump from one topic to the next one, which is very confusing.
It would very much improve their intensive work on this topic if authors revise the manuscript and improve discussion of their summarized data.
Minor points:
-Please consequently use aberrations
- Line 45: please specify which TNF (alpha?) and IL-1 (beta?) , IL17 you mean.
-Line 52: sentence is very long und message unclear, please improve
-Line 100: should one really call CD154 a cytokine?
-Line 252: it is not about arthritis, you also mention SLE even at the beginning, so please change subheading title
-Line 415: please start with specifying your most important conclusions with regard to your review
Author Response
Response to Reviewer #2
Lai et al summarize current knowledge about CD40-CD154 signaling in arthritis. To this end they explain cellular signaling pathways, summarize knowledge about murine models and clinical experience with CD40-CD154 blockers. Authors summarize a very impressive and huge amount of data with regard to this topic. However at some points they fail to really discuss the data and to come to conclusions or to generate hypothesis. In several paragraphs they only list recent studies and jump from one topic to the next one, which is very confusing. It would very much improve their intensive work on this topic if authors revise the manuscript and improve discussion of their summarized data.
Response: We thank Reviewer’s precious time on our report and the intention to improve the quality of our work. The reviewer’s helpful comments are well-received. We authors do recognize the potential inadequacy of this report as suggested by the reviewer which is in part because of its extensive coverage of the field as also mentioned by the reviewer. However, also because of this, we provided the most important messages regarding the raised topic and yet we tried our best to make the raised individual issues as connected as possible.
In past few days, we spent quite a lot of time to read the draft again and again and we are sorry to say that we could not find any eminent or very unacceptable disconnection among individual paragraphs. We assume that different opinions about writing styles by us authors and the reviewer may account for this. We authors highly appreciate any specific suggestions from the reviewer (it will be very helpful to specifically point out the disconnected paragraphs or the requirement of extensive revision in writing for certain parts of the draft) so that we authors can focus to correct them. Again, we thank and appreciate the reviewer’s helpful comments on our manuscript and would like to revise the draft if it is necessary.
Minor points:
Line 45: please specify which TNF (alpha?) and IL-1 (beta?) , IL17 you mean.
Response: The current anti-TNF regimens are blocking both TNF-alpha and TNF-beta. The regimen targeting IL-1 is the IL-1 receptor antagonist. For targeting IL-17, currently commercially available regimens are targeting IL-17A and other regimens such as targeting IL-17A and IL-17F are emerging and already undergoing phase III clinical trial (DOI: 10.1080/14712598.2019.1555235). Most general review articles did not explain these details, a manner also chosen by us, because these details may not be the major points of the review. Additional information about biological DMARDs has also been covered in our commentary recently published in Biochemical Pharmacology 2019.
Line 52: sentence is very long und message unclear, please improve
Response: This sentence has been re-written according to the suggestion (the changes are in red color).
Line 100: should one really call CD154 a cytokine?
Response: The concern has been corrected (the changes are in red color).
Line 252: it is not about arthritis, you also mention SLE even at the beginning, so please change subheading title
Response: This has been corrected (the changes are in red color).
Line 415: please start with specifying your most important conclusions with regard to your review
Response: Perspectives has been re-organized as suggested (the changes are in red color).
We authors thank again the reviewer very helpful comments on our report.